# Lewis acid-catalyzed asymmetric reactions of β, γ-unsaturated 2-acyl imidazoles

Tengfei Kang[1], Liuzhen Hou[1], Sai Ruan[1], Weidi Cao[1], Xiaohua Liu [1✉] & Xiaoming Feng [1✉]

The investigation of diverse reactivity of β,γ-unsaturated carbonyl compounds is of great value in asymmetric catalytic synthesis. Numerous enantioselective transformations have been well developed with β,γ-unsaturated carbonyl compounds as nucleophiles, however, few example were realized by utilizing them as not only nucleophiles but also electrophiles under a same catalytic system. Here we report a regioselective catalytic asymmetric tandem isomerization/α-Michael addition of β,γ-unsaturated 2-acyl imidazoles in the presence of chiral *N,N'*-dioxide metal complexes, delivering a broad range of optically pure 1,5-dicarbonyl compounds with two vicinal tertiary carbon stereocenters in up to >99% ee under mild conditions. Meanwhile, stereodivergent synthesis is disclosed to yield all four stereoisomers of products. Control experiments suggest an isomerization process involved in the reaction and give an insight into the role of NEt$_3$. In addition, Mannich reaction and sulfur-Michael addition of β,γ-unsaturated 2-acyl imidazoles proceed smoothly as well under the same catalytic system.

[1] Key Laboratory of Green Chemistry & Technology, Ministry of Education, College of Chemistry, Sichuan University, Chengdu 610064, China. ✉email: liuxh@scu.edu.cn; xmfeng@scu.edu.cn

The exploration of reaction diversity from β,γ-unsaturated carbonyl compounds is interesting and of great synthetic value. These compounds and their analogs bearing one potential enolization have been demonstrated as highly active nucleophiles in a number of catalytic asymmetric reactions for the synthesis of natural products and bioactive compounds[1–16]. Especially, γ-addition as dienolate pronucleophiles with either metal catalysis[17–28] or organocatalysis[29–36] has been widely documented during the past several years, and the maintained π-conjugation of γ-addition process leading to thermodynamically stable conjugated products (Fig. 1a, A). The regioselectivity changing from γ-addition to α-addition seems to be plaguing[37,38], and α-addition of specific substrates, such as γ,γ-disubstituted

ones, has been reported[39–43]. Notably, in some cases, C=C isomerization occurred after α-addition which further expanded the reaction diversity (Fig. 1a, B)[44–46].

Although versatile catalytic asymmetric reactions have been demonstrated by utilizing β,γ-unsaturated carbonyl compounds as mentioned above, however, few examples were investigated by employing them as electrophiles upon isomerization to conjugated α,β-unsaturated carbonyl compounds (Fig. 1a, C)[47,48]. We envision that, by careful design of β,γ-unsaturated carbonyl compounds, these could serve not only as nucleophiles but also electrophiles. Based on this assumption, here we report the synthesis of a series of β,γ-unsaturated 2-acyl imidazoles by introducing an imidazole moiety which would address the

**Fig. 1 Strategies for γ- and α-addition of β,γ-unsaturated carbonyl compounds. a** Regioselectivity of decojugated carbonyl compounds. **b** Our strategies for diverse reactivity of β,γ-unsaturated 2-acyl imidazoles.

following two points: (1) bidentate coordination with a Lewis acid of acyl imidazole exhibits good stereocontrol[49–55] and (2) the strong coordination facilitates isomerization of the β,γ-unsaturated ketone to an α,β-unsaturated ketone. Chiral $N,N'$-dioxide-metal[56–59] complexes catalyze diverse reactions of β,γ-unsaturated 2-acyl imidazoles, including tandem isomerization/α-Michael addition (Fig. 1b, D), Mannich reaction (Fig. 1b, B), and sulfur-Michael addition (Fig. 1b, C) with high efficiency and stereoinduction. In addition, stereodivergent catalysis[60–63] is also disclosed and provides a unified and predictable route for the access to all four stereoisomers of 1,5-dicorbonyl compounds by matching the configuration between the Lewis acid catalysts and substrates.

## Results

**Optimization of the reaction conditions**. We began our study by employing β,γ-unsaturated 2-acyl imidazole $E$-**1a** as the model substrate to optimize the reaction conditions. Several metal salts coordinated with the $N,N'$-dioxide ligand **L₃-RaPr₂** (Fig. 2) were evaluated, such as Sc(OTf)₃, Ni(OTf)₂, and Mg(OTf)₂; however, only trace amount of the self-α/β-addition product **2a** was observed, which was generated from α-addition of $E$-**1a** with the corresponding α,β-unsaturated 2-acyl imidazole upon C=C isomerization (Table 1, entry 1). Pleasingly, the Y(OTf)₃/**L₃-RaPr₂** complex was efficient to promote the tandem isomerization/α-Michael addition and provided the corresponding product **2a** with 60% yield, 2.2:1 *anti:syn* ratio, and 96% ee in CH₂ClCH₂Cl

(entry 2). Lanthanide metal salts La(OTf)₃ and Yb(OTf)₃ could also mediate the reaction but gave lower yields and ee values (entries 3 and 4). The screening of chiral backbones and steric hindrance of the amide moiety on the $N,N'$-dioxide ligands afforded no better results (for details, see Supplementary Table 1). When toluene was used as solvent instead, the isolated yield of anti-**2a** was increased to 73% with 5.2:1 dr and 97% ee (entry 5). To our delight, the diastereoselectivity could be improved to 10:1 with addition of NEt₃ (entry 6). Other common chiral ligands such as Box, Pybox, and BINAP were also explored, and 32% yield, 5:1 dr with 60% ee were observed as the best results (for details, see Supplementary Table 3).

**Substrate scope in isomerization/α-Michael addition reaction**. The generality of the tandem isomerization/α-Michael addition reaction was investigated under the optimized conditions (Fig. 3). An array of β,γ-unsaturated 2-acyl imidazoles bearing different substituents on the γ-phenyl group (both electron-withdrawing and electron-donating groups at the *para-*, *meta-*, or *ortho-* positions) were converted into the corresponding dimerization products **2a–2j** in good yields (65–81%), high diastereoselectivities (7.5:1 to 11:1), and excellent ee values (97–>99%). Furthermore, β,γ-unsaturated carbonyl compounds containing 3-thienyl, *N*-methyl-5-indolyl and 2-naphthyl moieties were also proven to be suitable substrates, affording **2k–2m** with good results (60–81% yields, 9:1 to 12:1 dr, and 98–>99% ee). Moreover, aliphatic-substituted β,γ-unsaturated 2-acyl imidazoles

**L₃-RaPr₂**: Ar = 2,6-*i*-Pr₂C₆H₃
**L₃-RaPr₂-1-Ad**: Ar = 2,6-*i*-Pr₂-4-(1-admantanyl)C₆H₂

**a**

**L₃-PePr₃**: Ar = 2,4,6-*i*-Pr₃C₆H₂

**b**

**L₃-Pi^tBu**

**c**

**Fig. 2 Representative chiral $N,N'$-dioxide ligands used in the study. a** L-Ramipril-derived ligand **L₃-RaPr₂** and **L₃-RaPr₂-1-Ad**. **b** L-Perindopril-derived ligand **L₃-PePr₃**. **c** S-pipecolic acid-derived ligand **L₃-Pi^tBu**.

---

### Table 1 Optimization of the reaction conditions.

| Entry | metal salt | Yield (%)[a] | *anti:syn*[b] | ee (%)[c] |
|---|---|---|---|---|
| 1 | Sc(OTf)₃/Ni(OTf)₂/Mg(OTf)₂ | Trace | — | — |
| 2 | Y(OTf)₃ | 60 | 2.2:1 | 96/−34 |
| 3 | La(OTf)₃ | 58 | 2.7:1 | 92/63 |
| 4 | Yb(OTf)₃ | 46 | 2.2:1 | 84/13 |
| 5[d] | Y(OTf)₃ | 73 | 5.2:1 | 97/0 |
| 6[d,e] | Y(OTf)₃ | 74 | 10:1 | 98/N.D. |

Unless otherwise noted, all reactions were performed with metal salt/ligand (1:1, 2.5 mol%), $E$-**1a** (0.20 mmol) in CH₂ClCH₂Cl (1.0 mL) at 25 °C under N₂ atmosphere for 24 h. [a]Isolated yield of *anti*-isomer. [b]Determined by ¹H NMR analysis of crude products. [c]Determined by HPLC analysis on a chiral stationary phases. [d]Toluene was used as solvent. [e]Addition of NEt₃ (10 mol%) and for 12 h.

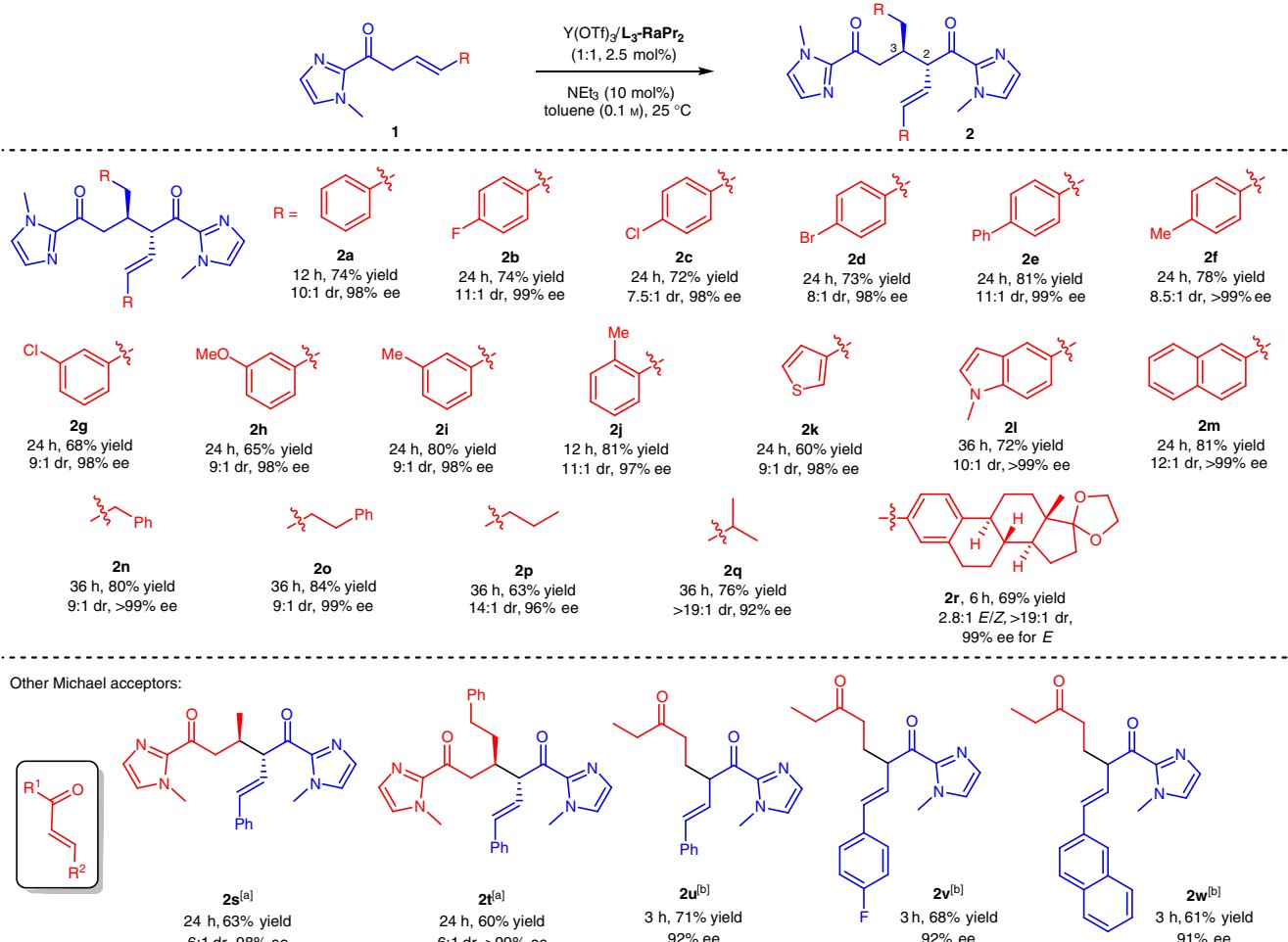

**Fig. 3 Substrate scope in isomerization/α-Michael addition reaction.** Unless otherwise noted, all reactions were performed with Y(OTf)$_3$/**L$_3$-RaPr$_2$** (1:1, 2.5 mol%), **1** (0.20 mmol), NEt$_3$ (10 mol%) in toluene (1.0 mL) at 25 °C under N$_2$ atmosphere. The yield was based on isolated *anti*-isomer. The dr value was determined by $^1$H NMR of crude products. The ee value was determined by HPLC analysis on chiral stationary phases. The substrates **1l** and **1n–1r** were used as *Z/E* mixutres. [a] 5 mol% catalyst was used for **2s** and **2t**. [b] With 5 mol% Y(OTf)$_3$/**L$_3$-RaPr$_2$-1-Ad** as a catalyst and CH$_2$Cl$_2$ as a solvent in the absence of NEt$_3$ for **2u–2w**.

exhibited high tolerance as well, generating the desired products **2n–2q** with a high level of yields (63–84%) and stereoselectivities (9:1 to >19:1 dr; 92–>99% ee). Estrone-derived **1r** could be transformed into **2r** smoothly in 69% yield, 2.8:1 *E/Z*, >19:1 dr, and 99% ee for *E*-isomer. Other Michael acceptors such as α,β-unsaturated 2-acyl imidazole and ethyl vinyl ketone were aslo suitable in this reaction, delivering **2s–2w** with good yields (60–71%) and stereoselectivities (6:1 dr, 91–>99% ee). The absolute configuration of **2j** was determined to be (2*S*, 3*R*) by X-ray crystallography analysis.

**Substrate scope in α-Mannich reaction of β,γ-unsaturated 2-acyl imidazoles and imines.** The reaction described above indicated that β,γ-unsaturated 2-acyl imidazoles performed both α-addition reaction and β-addition upon isomerization under proper Lewis acid catalysts. Next, to extend the scope of α-addition of β,γ-unsaturated 2-acyl imidazoles, several types of imines **3** were explored as the electrophiles. By switching the catalyst to La(OTf)$_3$/**L$_3$-Pi$^t$Bu** complex (for detailed screening of the conditions, see Supplementary Table 4), the Mannich reaction between *E*-**1** and isatin-derived ketimines **3a–3h** was successfully realized to deliver the desired β-amino 2-acyl imidazoles **4a–4h** as single isomers in 75–99% yields and 88–91% ee (Fig. 4a).

Moreover, pyrazolinone-derived ketimine was also suitable in this α-addition reaction, no matter β-aryl-substituted or β-alkyl-substituted β,γ-unsaturated 2-acyl imidazoles could react with it smoothly, producing the corresponding products **4i–4o** and **4q** with good results (75–99% yields, 13:1–>19:1 dr, 85–99% ee) except for **4p** with 52% ee (Fig. 4b). Aldimines were used as the Mannich acceptors, and were transformed into the β-amino 2-acyl imidazoles **4r–4z** with good yields (55–81%) and high enantioselectivities (85–98% ee) as single isomers (Fig. 4c). The absolute configuration of **4r** was determined to be (1*S*, 2*R*) by X-ray crystallography analysis.

**Substrate scope in isomerization/sulfur-Michael reaction.** Inspired by the isomerization process of β,γ-unsaturated 2-acyl imidazoles into α,β-unsaturated 2-acyl imidazoles, we next enlarged the diverse reactivity of β,γ-unsaturated compounds as the electrophiles under the current catalytic system. However, only a trace amount of desired tandem isomerization/sulfur-Michael addition product **6a** was achieved if *E*-**1a** reacted with thiophenol **5a**. After examination of the reaction conditions (for details, see Supplementary Table 5), *Z*-**1a** was used instead, and **6a** could be obtained in 89% yield with 90% ee (Fig. 5). The scope of isomerization/sulfur-Michael reaction was investigated next.

**L = L₃-Pi^tBu, 3 Å M.S., CH₂ClCHCl₂, 0 °C**

**4a**, R¹ = H, R = Ph, 12 h, 95% yield, >19:1 dr, 91% ee
**4b**, R¹ = 5-Cl, R = Ph, 5 h, 87% yield, >19:1 dr, 90% ee
**4c**, R¹ = 5-I, R = Ph, 6 h, 90% yield, >19:1 dr, 90% ee
**4d**, R¹ = 5-Me, R = Ph, 12 h, 75% yield, >19:1 dr, 90% ee
**4e**, R¹ = 6-Cl, R = Ph, 16 h, 88% yield, >19:1 dr, 90% ee
**4f**, R¹ = 6-F, R = Ph, 24 h, 98% yield, 19:1 dr, 88% ee
**4g**, R¹ = 7-Br, R = Ph, 16 h, 95% yield, 19:1 dr, 88% ee
**4h**, R¹ = H, R = 2-naphthyl, 36 h, 99% yield, 19:1 dr, 88% ee

**L = L₃-RaPr₂, CHCl₃, 0 °C**

**4i**, R = Ph, 8 h, 94% yield, >19:1 dr, 99% ee
**4j**, R = 4-BrC₆H₄, 12 h, 98% yield, 14:1 dr, 98% ee
**4k**, R = 3-MeC₆H₄, 10 h, 99% yield, 16:1 dr, 96% ee
**4l**, R = 3-MeOC₆H₄, 8 h, 97% yield, >19:1 dr, 98% ee
**4m**, R = 2-MeC₆H₄, 10 h, 98% yield, >19:1 dr, 98% ee
**4n**, R = 3-thienyl, 8 h, 99% yield, >19:1 dr, 98% ee
**4o**[a], R = Bn, 10 h, 96% yield, 13:1 dr, 85% ee
**4p**[a], R = propyl, 4 h, 97% yield, 10:1 dr, 52% ee

**4q**[a]
10 h, 75% yield
>19:1 dr, 99% ee

**L = L₃-RaPr₂, CH₂ClCHCl₂, 4 Å M.S., 35 °C**

**4r**, R = Ph, R² = Ph, 10 h, 73% yield, >19:1 dr, 98% ee
**4s**, R = Ph, R² = 4-ClC₆H₄, 10 h, 77% yield, >19:1 dr, 97% ee
**4t**, R = Ph, R² = 3-MeC₆H₄, 10 h, 70% yield, >19:1 dr, 98% ee
**4u**, R = Ph, R² = 3-BrC₆H₄, 8 h, 57% yield, >19:1 dr, 95% ee
**4v**, R = Ph, R² = 2-MeC₆H₄, 8 h, 55% yield, >19:1 dr, 97% ee
**4w**, R = 2-MeC₆H₄, R² = Ph, 8 h, 67% yield, >19:1 dr, 96% ee
**4x**, R = 3-MeOC₆H₄, R² = Ph, 8 h, 62% yield, >19:1 dr, 98% ee
**4y**, R = 4-BrC₆H₄, R² = Ph, 8 h, 72% yield, >19:1 dr, 97% ee

**4z**
10 h, 81% yield
>19:1 dr, 85% ee

**Fig. 4 Substrate scope in α-Mannich reaction of β,γ-unsaturated 2-acyl imidazoles and imines. a** Substrate scope with isatin-derived ketimins. **b** Substrate scope with pyrazolinone-derived ketimins. **c** Substrate scope with aldimines. Unless otherwise noted, all the reactions were performed with La(OTf)₃/**Ligand** (1:1, 5 mol%), **1** (0.10 mmol), **3** (0.10 mmol for **4a**–**4q**, 0.15 mmol for **4r**–**4z**) in the indicated solvent. The dr value was determined by ¹H NMR of crude products. The ee value was determined by HPLC analysis on chiral stationary phases. [a] At 20 °C. [b] With 10 mol% of catalyst.

Thiolphenols and alkyl-substituted thiols could be converted into the final products (**6a–6i**) in 39–95% yields with 70–93% ee values. For the Michael acceptors, aryl- and alkyl-substituted β,γ-unsaturated 2-acyl imidazoles were also tolerated in this reaction, giving **6j–6p** in 60–92% yields with 80–92% ee.

**Gram-scale synthesis and derivatization of products**. To evaluate the synthetic utility of this methodology, a gram-scale synthesis of **2a** was conducted. The current reaction could be carried out at 7.0 mmol scale without loss of yield (70%), diastereoselectivity (10:1 dr), and ee value (98%) (Fig. 6a). Furthermore, hydrogenation of **2a** in the presence of Pd/C and H₂ afforded derivative **7** in 98% yield with 98% ee (Fig. 6b). Chiral sulfone motif is found in numerous biological compounds[64–67] as well as drug candidates[68]. Upon treatment of **6a** with m-CPBA, the oxidized sulfone product **8** was obtained in 85% yield with

90% ee. Moreover, **6a** went through further transformations to afford sulfone **9** in 50% yield with 85% ee (Fig. 6c)[69].

**Mechanistic studies**. To gain insight into the mechanism of tandem isomerization/α-Michael addition, some control experiments were carried out. Firstly, we wondered why the addition of NEt₃ led to an increase in diastereoselectivity (Table 1, entry 6). Treating the product **2a** (2.9:1 dr, 85%/12% ee) under the standard conditions for 12 h (for details, see Supplementary Note 5), no change of enantioselectivity and diastereoselectivity was observed, which ruled out the possibility that the diastereoselectivity increased via epimerization of syn-**2a** in the presence of NEt₃. Subsequently, E-α,β-unsaturated 2-acyl imidazole E-**10** was synthesized to react with E-**1a**, affording anti-**2a** in good yields (84–85%), excellent diastereoselectivities (19:1 to >19:1), and 98% ee within 2 h no matter with or without addition of NEt₃ (Fig. 7a).

**Fig. 5 Substrate scope in isomerization/sulfur-Michael reaction.** Unless otherwise noted, all reactions were performed with Dy(OTf)$_3$/**L$_3$-PePr$_3$** (1:1, 5 mol%), **1** (0.25 mmol), **5** (0.10 mmol) in CH$_2$ClCHCl$_2$ (1.0 mL) at 25 °C for 17 h. [a] Z/E mixture of β,γ-unsaturated 2-acyl imidazole was used for **6n**. The reaction time was 5 days.

**Fig. 6 Gram-scale synthesis and derivatization of products. a** Gram-scale synthesis of **2a**. **b** Hydrogenation of **2a**. **c** The derivatization of **6a**, (1) PhMgBr, THF; (2) MeOTf, MeCN; (3) K$_2$CO$_3$ (aq); (4) m-CPBA, CH$_2$Cl$_2$.

Moreover, when Z-α,β-unsaturated 2-acyl imidazole Z-**10** was used to react with E-**1a**, the product **2a** was obtained in 1:5.2 anti: syn after 2 h, and decreased to 1:2.8 anti:syn after 5 h (Fig. 7b). These experiments confirmed the isomerization of β,γ-unsaturated C=C bond into α,β-unsaturated C=C bond in the presence of N,N′-dioxide-metal complexes, and this process was likely to be the rate-determining step. It also suggests the diastereoselectivity was mainly controlled by the E/Z-configuration of the α, β-unsaturated 2-acyl imidazole intermediate, and the addition of

NEt$_3$ might improve the E/Z ratio during the isomerization process. As a result of equilibrium between E-**1a**, E-**10**, and Z-**10** (Fig. 7c), the use of E-**10** as the starting substrate alone, albeit unstable yielded the corresponding anti-**2a** as the major product in 98% ee after 3 h (Fig. 7d), while the reaction from only Z-**10** gave the syn-**2a** product in 60% isolated yield and 92% ee (Fig. 7e). In addition, operando IR experiments were also performed to interpret the reaction process (for details, see Supplementary Note 7). Furthermore, we set out to establish the

**Fig. 7 Mechanistic studies. a** Reaction of *E*-**10** with *E*-**1a**. **b** Reaction of *Z*-**10** with *E*-**1a**. **c** Isomerization of *E*-**1a** with *E*-**10** and *Z*-**10**. **d** Reaction of single *E*-**10**. **e** Reaction of single *Z*-**10**. **f** Stereodivergent synthesis of **2a**. [a] *m*-Xylene was used instead of toluene.

availability of stereodivergent access to **2a**. All four stereoisomers of **2a** could be readily obtained in good yields (67–85%) and diastereoselectivities (8:1–>19:1) with excellent ee values by matching the *E*/*Z*-configurated **10** and the chiral ligand (Fig. 7f).

**Proposed catalytic cycle.** Based on the absolute configuration of the product **2j**, control experiments and our previous studies[56–59], a possible catalytic cycle with a transition-state model was proposed (Fig. 8). First, the coordination of chiral *N*,*N*′-dioxide **L₃**-**RaPr₂** and metal salt in situ to form chiral metal complex (**Y***). Then, the β,γ-unsaturated ketone *E*-**1a** attaches to **Y*** as a dienolate in the presence of NEt₃ to give the intermediate **T1**, and which partly transforms into the α,β-unsaturated ketone *E*/*Z*-**10** upon 1,5-proton shift. Next, the catalyst-bonded dienolate will react with the newly formed Michael acceptors. The α-*Re*-face of β,γ-unsaturated 2-acyl imidazole *E*-**1a** is strongly shielded by the nearby aryl ring of the ligand. Therefore, the dienolate prefers to attack *E*/*Z*-**10** from its α-*Si*-face (**T2**). Finally, the desired product

**2a** dissociates after a protonation of the intermediate **T3**, and the catalyst is regenerated to accomplish one catalytic cycle.

## Discussion

In summary, we have disclosed the diverse transformation of β,γ-unsaturated 2-acyl imidazoles in the presence of chiral Lewis acid catalysts, involving catalytic asymmetric tandem isomerization/α-Michael addition, sulfur-Michael addition, and direct Mannich reaction. A wide range of chiral 1,5-dicarbonyl and functionalized carbonyl compounds was afforded with good to excellent levels yields, diastereoselectivities, and enantioselectivities. The β,γ-unsaturated 2-acyl imidazoles features various reactivities, acting as both α-nucleophile and β-electrophile upon isomerization, which provides a route for conjugate addition of unstable α,β-unsaturated carbonyl compounds. Meanwhile, all four stereoisomers with two vicinal tertiary stereocenters could be prepared by matching the configuration between substrates and chiral ligand. Besides, the desired products could be easily transformed

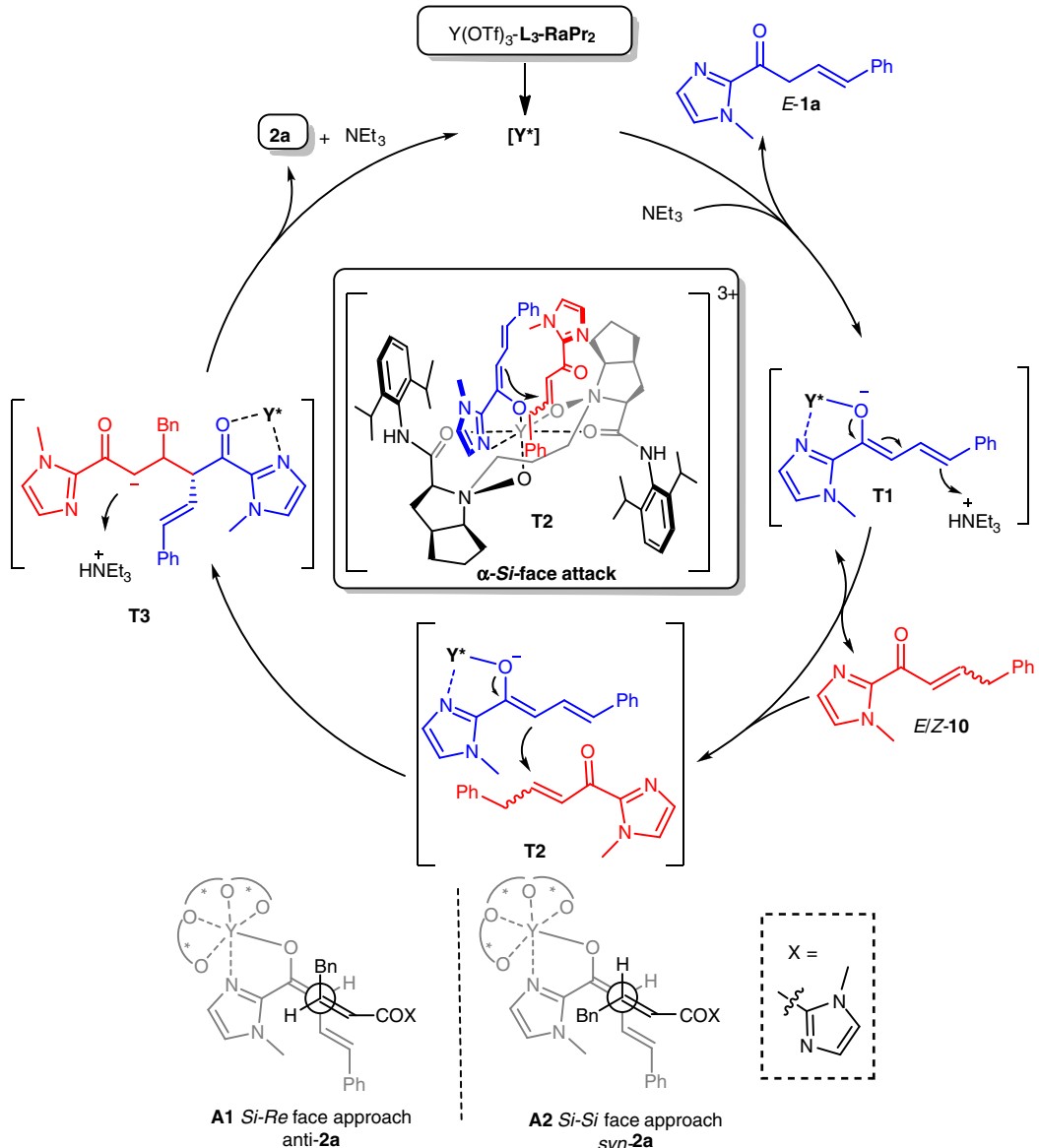

**Fig. 8 Proposed catalytic cycle.** The in situ formed chiral catalyst [**Y\***] catalyzes isomerization of *E*-**1a** into *E/Z*-**10** in the presence of NEt₃, followed by nucleophilic addition of *E*-**1a** and protonation to deliver the final product **2a**.

into useful compounds with good results under mild conditions. Further studies on this methodology are ongoing.

## Methods

**Tandem isomerization/α-Michael addition**. Y(OTf)₃ (0.005 mmol), **L₃-RaPr₂** (0.005 mmol), β,γ-unsaturated 2-acyl imidazole *E*-**1a** (0.20 mmol), and NEt₃ (0.02 mmol) were dissolved in 1.0 mL of toluene under N₂ atmosphere. The mixture was stirred at 25 °C for 12 h and subjected to column chromatography on silica to afford the product **2a** (Pet/EtOAc = 1:1 as eluent) as a colorless foam.

**Mannich reaction with isatin-derived ketimines**. A dry reaction tube was charged with **L₃-Pi^tBu** (2.2 mg, 5 mol%), La(OTf)₃ (2.9 mg, 5 mol%), 3 Å M.S. (30 mg), and *E*-**1a** (27.1 mg, 0.12 mmol) in CH₂ClCHCl₂ (1.0 mL). The mixture was stirred at 30 °C for 30 min, and then **3a** (0.10 mmol, 26.0 mg) was added at 0 °C. After **3a** was consumed (detected by thin-layer chromatography (TLC)), the residue was purified by column chromatography on silica gel to afford the product **4a** (Pet/EtOAc = 1:1 as eluent) as a colorless foam.

**Mannich reaction with pyrazolinone-derived ketimines**. A dry reaction tube was charged with **L₃-RaPr₂** (3.5 mg, 5 mol%), La(OTf)₃ (2.9 mg, 5 mol%), *E*-**1a** (24.9 mg, 0.11 mmol), and pyrazolinone-derived ketimine (34.9 mg, 0.10 mmol) in

CHCl₃ (1.0 mL). After ketimine was consumed (detected by TLC), the residue was purified by column chromatography on silica gel to afford the product **4i** (Pet/EtOAc = 2:1 as eluent) as a colorless foam.

**Mannich reaction with aldimines**. A dry reaction tube was charged with **L₃-RaPr₂** (7.0 mg, 10 mol%), La(OTf)₃ (5.9 mg, 10 mol%), *E*-**1a** (24.9 mg, 0.10 mmol), 4 Å M.S. (20 mg), and benzaldehyde-derived aldimine (30.8 mg, 0.15 mmol) in CH₂ClCHCl₂ (1.0 mL). After *E*-**1a** was consumed (detected by TLC), the residue was purified by column chromatography on silica gel to afford the product **4r** (Pet/EtOAc = 2:1 as eluent) as a colorless oil.

**Isomerization/sulfur-Michael reaction**. A dry reaction tube was charged with **L₃-PePr₃** (4.2 mg, 5 mol%), Dy(OTf)₃ (3.0 mg, 5 mol%), and *Z*-**1a** (56.5 mg, 0.25 mmol) in CH₂ClCHCl₂ (1.0 mL). PhSH (0.10 mmol) was added and the mixture was stirred at 25 °C for 17 h. After PhSH was consumed (detected by TLC), the residue was purified by column chromatography on silica gel to afford the product **6a** (Pet/EtOAc = 3:1 as eluent) as a pale yellow oil.

## Data availability

The X-ray crystallographic coordinates for structures reported in this study have been deposited at the Cambridge Crystallographic Data Centre (CCDC), under deposition numbers CCDC 1972987 (**2j**), 2001513 (**4r**), and 1972937 (**11**). These data can be

obtained free of charge from The Cambridge Crystallographic Data Centre via https://www.ccdc.cam.ac.uk/data_request/cif. All other data are available from the corresponding author upon reasonable request.

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

## Acknowledgements
We appreciate the National Natural Science Foundation of China (Nos. 21890723 and 21921002) for financial support. Thanks Dr. Yuqiao Zhou for the assistance in X-ray analysis.

## Author contributions
T.K. performed experiments and prepared the Supplementary Information and paper. L.H. took part in the reaction development and synthesized several substrates. S.R. repeated some experiments. W.C. and X.L. helped with modifying the paper and Supplementary Information. X.F. conceived and directed the project.

## Competing interests
The authors declare no competing interests.
