## [Peer Review File · Nature Communications]

Reviewers' comments:

Reviewer #1 (Remarks to the Author):

Feng and co-workers report three types of enantioselective reactions (isomerization/ α -Michael addition reaction, Mannich reaction, and sulfur-Michael addition reaction) with the same type of substrates, β,γ -unsaturated 2-acyl imidazoles, by Lewis acid catalysis. In these transformations, a chiral chiral N,N'-dioxide ligand the authors have used for the other reactions in their previous works has been utilized. A series of chiral products have been achieved in good yields with high enantio- and diastereoselectivities. The authors have conducted control experiments to elucidate the reaction mechanisms, and then pointed out that when using Z- or E-substrates, four stereoisomers of 1,5-dicarbonyl compounds could be selectively formed under the similar conditions. This work is interesting. Although catalytic enantioselective reactions with olefinic 2-acyl imidazoles and the analogues have been reported, this work shows the diverse reactivity of olefinic 2-acyl imidazoles can be accomplished just simply adjusting the reaction conditions. This work is also inspiring for the people who are working on Lewis acid catalysis and the olefinic ketone chemistry. Furthermore, this manuscript is well prepared and the Supplementary Information is fine. This referee considers this manuscript can be accepted after the following two issues are addressed:

(1) Is substrate 1o or 1p suitable for the Mannich reaction and sulfur-Michael reaction? If yes, add the examples in the corresponding Schemes.

(2) In Scheme 5, E-10 gives the product in better diastereoselectivity than Z-10. In Scheme 6, how to explain this result on the basis of transition state T2.

Reviewer #2 (Remarks to the Author):

A regioselective catalytic Michael addition is shown using β,γ unsaturated 2-acyl imidazoles as substrates and chiral metal complexes as catalysts. The substrate partly isomerizes during the reaction to the more stable conjugated isomer which is probably attacked by a dienolate formed from the substrate by aid of the catalytic system and a base. In that way anti configured 1,5-dicarbonyl compounds with 2 adjacent stereocenters were formed. It is also shown that the syn isomer is available with low to moderate diastereoselectivity starting from the isolated Z configured conjugated isomer. Moreover, Mannich reactions to isatin derivatives proceeded with good stereoselectivities, whereas sulfa-Michael additions to the α,β unsaturated 2-acyl imidazoles proceeded with only moderate to good enantioselectivity.

Overall this manuscript reports an interesting advance, but one that does not pass the bar for publication in Nat. Commun. The scope of the first two reactions is very much limited and enantioselectivities are not always very high. To be more precise, the Michael addition is limited to a homocoupling (dimerization), the Mannich addition is limited to just isatin derived imines, and the sulfa Michael addition is limited to a single Michael acceptor and provided in 4 out of 7 cases ee values lower than 90%. Moreover, the mechanistic studies are not compelling and are mainly limited to some control experiments.

In its current form, this manuscript is better suited for a specialized journal like Adv. Synth. Cat. but with a more detailed mechanistic analysis and expanded scope, it would qualify as a full paper in a journal like Chem. Eur. J.

Reviewer #3 (Remarks to the Author):

This is a very beautiful work from Prof. Feng's group on the diverse reactivity on β,γ -unsaturated carbonyl compounds. The authors report a regioselective catalytic asymmetric tandem isomerization/ α -Michael addition of β,γ -unsaturated 2-acyl imidazoles in the presence of chiral N,N'-dioxide metal complexes they developed, delivering chiral 1,5-dicarbonyl compounds with

two vicinal tertiary carbon stereocenters in up to >99% ee. Meanwhile, stereodivergent synthesis was obtained to produce all four stereoisomers of products. In addition, the α -Mannich reaction of β,γ -unsaturated 2-acyl imidazoles, and sulfur-Michael addition of β,γ -unsaturated 2-acyl imidazoles were also developed. Interesting mechanistic information was disclosed. Thus, I strongly recommend this manuscript to be published in Nature Communication. I am curious about the following possibility:

1. Is it possible to improve syn-diastereoselectivity when β,γ -unsaturated 2-acyl imidazoles were used as α -selective Michael donors?
2. Did the authors try other Michael acceptors when β,γ -unsaturated 2-acyl imidazoles acted as α -selective Michael donors ?

Anyway, this is a very interesting work which is worth to be published in NC.

Reply to the comments by reviewer 1

1. **Question:** Is substrate **1o** or **1p** suitable for the Mannich reaction and sulfur-Michael reaction? If yes, add the examples in the corresponding Schemes.

Reply: Substrate **1p** is also suitable for the Mannich reaction and sulfur-Michael reaction, but with lower enantioselectivities. The corresponding results were listed as below and added into the revised manuscript (Scheme 3b and Scheme 4).

2. **Question:** In Scheme 5, *E*-**10** gives the product in better diastereoselectivity than *Z*-**10**. In Scheme 6, how to explain this result on the basis of transition state **T2**.

Reply: As the result that the *Z*-**10** affords syn-product, *E*-**10** affords anti-product (on the basis of transition state **T2**), in theory, high syn-diastereoselectivity will be obtained through the reaction of *E*-**1a** and *Z*-**10**, however, *Z*-**10** is unstable and easily transforms into *E*-**1a**, and *E*-**1a** tends to transform into *E*-**10** prior to *Z*-**10** (see below). Thus, the syn-diastereoselectivity will be high at the beginning of the reaction, but it will decrease as the conversion of *Z*-**10** to *E*-**10** during the reaction process.

Reply to the comments by reviewer 2

1. **Question:** The scope of the first two reactions is very much limited and enantioselectivities are not always very high. To be more precise, the Michael addition is limited to a homocoupling (dimerization), the Mannich addition is limited to just isatin derived imines, and the sulfa Michael addition is limited to a single Michael acceptor and provided in 4 out of 7 cases ee values lower than 90%.

Reply: Based on the reviewer's suggestion, the substrate scopes of Michael addition, Mannich addition and sulfa Michael addition were expanded, the results were updated in the revised manuscript (Scheme 2, 3 and 4). **For the Michael addition**, different Michael

acceptors were suitable in the isomerization/ α -Michael addition reaction, including dimerization, α,β -unsaturated 2-acyl imidazole and ethyl vinyl ketone. The enantioselectivities were excellent (>90% ee for all the corresponding products) (Scheme 2). **For the Mannich addition**, the substrate scope was expanded to pyrazolinone-derived ketimine and aldimines besides isatin derived imines, the ee values of most examples (20 out of 26) were excellent (>90% ee) (Scheme 3). **For the sulfa-Michael addition**, the substrate scope of Michael acceptor was expanded (**6j-6p**), and the ee values of most examples were good to excellent (Scheme 4).

Scheme 2 Substrate scope in isomerization/ α -Michael addition reaction

Scheme 2. Unless otherwise noted, all reactions were performed with $Y(OTf)_3/L_3-RaPr_2$ (1:1, 2.5 mol%), **1** (0.20 mmol), NEt_3 (10 mol%) in toluene (1.0 mL) at 25 °C under N_2 atmosphere. The yield was based on isolated *anti*-isomer. The dr value was determined by 1H NMR. The ee value was determined by HPLC analysis on chiral stationary phases. The substrates **1l**, **1n-1r** were used as *Z/E* mixtures. [a] 5 mol% Catalyst was used for **2s** and **2t**. [b] With 5 mol% $Y(OTf)_3/L_3-RaPr_2-1-Ad$ as catalyst and CH_2Cl_2 as solvent in the absence of NEt_3 for **2u-2w**.

Scheme 3. Substrate scope in α -Mannich reaction of β,γ -unsaturated 2-acyl imidazoles and imines^[a]

[a] Unless otherwise noted, all reactions were performed with La(OTf)₃/Ligand (1:1, 5 mol%), **1** (0.10 mmol), **3** (0.10 mmol for **4i-4q** and **4A**, 0.15 mmol for **4r-4z**) in the corresponding solvent. The dr value was determined by ¹H NMR. The ee value was determined by HPLC analysis on chiral stationary phases. [a] At 20 °C. [b] With 10 mol% catalyst.

Scheme 4. Substrate scope in isomerization/ sulfur-Michael reaction.

Unless otherwise noted, all reactions were performed with Dy(OTf)₃/L₃-PePr₃ (1:1, 5 mol%), **1** (0.25 mmol), **5** (0.10 mmol) in CH₂ClCHCl₂ (1.0 mL) at 25 °C for 17 hours. *Z/E* Mixture of β,γ-unsaturated 2-acyl imidazole was used as substrate for **6n**.

2. Question: Moreover, the mechanistic studies are not compelling and are mainly limited to some control experiments.

Reply: To gain further insight into the mechanism, operando IR experiments were performed to interpret the process of the reaction, and the corresponding results have been added into the SI (Page 11). As depicted in Figure S1, **1a**, **2a** and **int.** were monitored by the operando IR spectrometer. The peaks at 968 cm⁻¹ is related to **1a** gradually decreased in intensity and the peaks at 854 cm⁻¹ is related to the intermediate **int.** gradually increased and then decreased in intensity. It was shown clearly that the amount of product **2a** (peak at 881 cm⁻¹) increased with the decrease of starting material. In Figure S2, it was found that the reaction rate was slow in the initial 2 h, and became faster as the formation of intermediate **int.**

Figure S1. The operando IR experimen

Figure S2. The trend of each component (X axes: reaction time; Y axes: absorbance unit). **1a**: peak at 968 cm⁻¹; **2a**: peak at 881 cm⁻¹; intermediate **int.**: peak at 854 cm⁻¹.

Reply to the comments by reviewer 3

1. **Question:** Is it possible to improve syn-diastereoselectivity when β,γ -unsaturated 2-acyl imidazoles were used as α -selective Michael donors?

Reply: As the result that the *Z*-**10** affords syn-product, *E*-**10** affords anti-product, in theory, high syn-diastereoselectivity will be obtained through the reaction of *E*-**1a** and *Z*-**10**, however, *Z*-**10** is unstable and easily transforms into *E*-**1a**, and *E*-**1a** tends to transform into *E*-**10** prior to *Z*-**10** (see below). Thus, the syn-diastereoselectivity will be high at the beginning of the reaction, but it will decrease as the conversion of *Z*-**10** to *E*-**10** during the reaction process.

After screening of some other conditions, the syn-diastereoselectivity of (*2S,3S*)-**2a** could be improved to 8:1 dr from previous 5.2:1 dr (entry 9), and the syn-diastereoselectivity of (*2R,3R*)-**2a** could be improved to 9:1 dr from previous 6.1:1 dr (entry 14). These results were revised in the new version (Scheme 6 in the manuscript).

Entry ^[a]	metal salt	ligand	solvent	time (h)	yield (%) ^[b]	dr ^[c]	ee (%) ^[d]
1	Y(OTf) ₃	L₃-RaPr₂	toluene	1	53	6.2:1	99
2 ^[e]	Y(OTf) ₃	L₃-RaPr₂	toluene	2	48	2.6:1	97
3 ^[f]	Y(OTf) ₃	L₃-RaPr₂	toluene	2	57	3.4:1	98
4	Sc(OTf) ₃	L₃-RaPr₂	toluene	2	<5	-	-
5	La(OTf) ₃	L₃-RaPr₂	toluene	2	56	8:1	99
6	Dy(OTf) ₃	L₃-RaPr₂	toluene	2	55	4.7:1	98
7	Yb(OTf) ₃	L₃-RaPr₂	toluene	2	40	3:1	97
8	La(OTf) ₃	L₃-RaPr₂	m -xylene	2	50	8.1:1	>99
9	Y(OTf) ₃	L₃-RaPr₂	m -xylene	2	67	8:1	>99
10	Y(OTf) ₃	L₃-RaPr₂	p -xylene	2	38	3.5:1	>99
11	Y(OTf) ₃	L₃-RaPr₂	Mesitylene	2	trace	-	-
12	Y(OTf) ₃	L₃-RaPr₂	Chlorobenzene	2	68	2.4:1	98
13	Y(OTf) ₃	L₃-RaPr₂	THF	2	63	4.9:1	98

[a] Unless otherwise noted, all reactions were performed with metal salt/Ligand (1:1, 5 mol%), *E*-1a (0.10 mmol), *Z*-10 (0.10 mmol) in solvent (1.0 mL) at 25 °C under N₂ atmosphere. [b] The yield was based on isolated *syn*-isomer. [c] The dr value was determined by ¹H NMR. [d] The ee value was determined by HPLC analysis on chiral stationary phases. [e] 10 mol% of catalyst was used. [f] Toluene (0.2M) was used.

2. **Question:** Did the authors try other Michael acceptors when β,γ-unsaturated 2-acyl imidazoles acted as α-selective Michael donors ?

Reply: We have explored a number of other Michael acceptors isomerization/α-Michael addition reaction (see below). The successful examples, such as α,β-unsaturated 2-acyl imidazole and ethyl vinyl ketone were added into the Scheme 2 in the revised manuscript. Some other failed examples were added in the revised Supporting Information (page 8).

Other failed examples of Michael acceptors in isomerization/α-Michael addition reaction:

We hope that the revised version is satisfactory for publication in *Nature communications*.
Thanks again for your assistance with the manuscript.

Best regards and look forward to hearing from you.

Sincerely yours,

Xiaoming Feng

REVIEWERS' COMMENTS:

Reviewer #1 (Remarks to the Author):

The authors well addressed the issues. This referee has no more questions and recommends acceptance.

Reviewer #2 (Remarks to the Author):

The authors have enormously increased the scope and utility of the three investigated reaction types. In addition careful spectroscopic studies have been performed to permit a better mechanistic understanding. Because the scientific quality of this paper has now been massively improved, I feel able to support publication of the revised manuscript.

Reviewer #3 (Remarks to the Author):

I am satisfactory with the reply on my comments, and I recommend this work for publication in Nature communications.